

# 3D-printed resin for permanent/definitive restorations: meta-analysis for bond strength

Minjoo Ki[1], Marc Hayashi[1,2] and Mijoo Kim[1,2]

[1] UCLA Biomaterials and Device Testing Laboratory, School of Dentistry, University of California, Los Angeles, Los Angeles, CA, United States of America
[2] Section of Restorative Dentistry, UCLA School of Dentistry, University of California, Los Angeles, Los Angeles, CA, United States of America

## ABSTRACT

**Background.** Three-dimensional (3D) printing technologies have gained increasing popularity in restorative dentistry for fabricating polymer-based restorations. However, limited evidence exists regarding the bond strength between 3D-printed restorative materials and resin cements, particularly considering the effects of surface treatments and aging conditions. This meta-analysis aimed to synthesize the current knowledge on the adhesion of 3D-printed restorations to resin cements and identify areas for future research.

**Methodology.** A literature search was conducted in PubMed, Google Scholar, and Web of Science databases using the following keywords: "3D - printing", "additive manufacturing", "bond strength", "resin cement", "surface treatment", and "aging". Inclusion criteria were studies evaluating the bond strength of 3D-printed polymer restorative materials to resin cements, published in English, and with full-text availability. Data on bond strength values, failure modes, surface treatments, aging protocols, and material characterization were extracted and analyzed.

**Results.** Nine studies were included in the review. Digital light processing (DLP) and stereolithography (SLA) were the predominant 3D printing technologies. Airborne-particle abrasion (APA) with 50 μm aluminum oxide particles was used in five studies and significantly improved bond strengths compared to untreated controls. Chemical treatments such as silane coupling agents and universal adhesives containing 10-MDP were used in some studies and further enhanced adhesion when combined with APA. Thermocycling for 5,000 cycles, simulating 6 months of clinical service, was the most common aging protocol. Bond strengths generally decreased after aging, with some material and surface treatment combinations showing greater stability than others.

**Conclusions.** APA is an effective surface treatment method for improving the bond strength of 3D-printed polymer restorations to resin cement. The combination of mechanical treatment with chemical agents such as silane and 10-MDP provides additional benefits. Material composition, particularly the use of UDMA-based resins, significantly influences bond stability after aging. Standardized protocols for 3D-printing, post-processing, and testing methods are essential for consistent results. Further clinical investigations are needed to establish long-term performance guidelines and optimize bonding protocols for these innovative restorative materials.

Corresponding author
Mijoo Kim,
mijookim@dentistry.ucla.edu

## INTRODUCTION

The adoption of 3D printing technology in dentistry has undergone a remarkable evolution, initially establishing itself through applications in provisional restorations, surgical guides, and dental splints. These early applications demonstrated the technology's capacity for precise customization and efficient production, particularly in creating temporary dental solutions and surgical planning tools (*Khorsandi et al., 2021*). Simultaneously, the potential of 3D printing in fabricating removable dentures attracted significant attention from developers and clinicians, as the technology promised to streamline the traditionally labor-intensive denture manufacturing process while maintaining precise fit and allowing easy duplication (*Anadioti et al., 2020*). As the technology matured and materials science advanced, interest grew in expanding the application of 3D-printed resins beyond provisional uses toward fixed permanent dental restorations, driven by the potential for improved workflow efficiency and reduced production costs (*Alghauli & Alqutaibi, 2024*).

The use of 3D-printed resin for permanent dental restorations represents a significant advancement in restorative dentistry, with bond strength being a critical factor in their long-term clinical success. While traditional methods like milling or casting have established protocols for achieving reliable bonds, the unique composition and manufacturing process of 3D-printed resins present new challenges for achieving optimal adhesion (*Poker et al., 2024*; *Tzanakakis et al., 2023*). The layer-by-layer fabrication process, while offering superior customization and efficiency, introduces variables that may affect the material's bonding characteristics (*Vanaei et al., 2020*; *Xiao et al., 2024*). High-filler 3D-printed materials, such as those with lithium disilicate and zirconia fillers, require specific surface treatments to achieve optimal bond strength compared to the traditional indirect restorations, mainly composed of ceramics (*Kim et al., 2024*). The surface topography of 3D-printed materials, influenced by the printing technology used, affects their bonding strength with adhesives. Different 3D-printed materials exhibit varying peel bond strengths with elastomeric impression systems, indicating that surface characteristics play a crucial role in bonding efficacy (*Xu et al., 2020*). The presence of various additives in polymeric resins, such as antioxidants, stabilizers, and pigments, presents another consideration, as these components can migrate and potentially affect the material's surface properties and bonding behavior over time.

Recent developments in 3D-printed materials have led to the introduction of high-strength biocompatible resins containing over 50% ceramic fillers, which are now approved for permanent restorations in the United States as of 2023 (*Zhao et al., 2021*). While these materials offer improved mechanical properties, their high filler content presents unique challenges for achieving optimal bond strength. The interaction between the resin matrix, ceramic fillers, and various surface treatments needs careful consideration to ensure reliable

adhesion (*Alghauli & Alqutaibi, 2024*; *Trembecka-Wójciga & Ortyl, 2024*). The proportion of fillers influences not only mechanical properties but also surface characteristics and bonding behavior, potentially requiring customized surface treatment protocols. Furthermore, the effectiveness of different surface treatments, such as sandblasting, etching, or silanization, may vary significantly compared to traditional restorative materials (*Ersöz et al., 2024*; *Kang et al., 2023*).

The long-term stability of adhesive bonds in 3D-printed restorations remains a critical concern, particularly given the material's susceptibility to aging effects and the dynamic oral environment (*Frassetto et al., 2016*). Factors such as thermal cycling, moisture exposure, and mechanical stress can potentially compromise the bond strength over time (*Bedran-De-Castro et al., 2004*). Additionally, the post-processing procedures unique to 3D-printed resins, including washing and post-curing protocols, may significantly influence their bonding characteristics (*Jin et al., 2022*).

Given these considerations, a meta-analysis evaluation of bond strength in 3D-printed permanent resins is essential for understanding the factors that influence their adhesive performance. This review aims to analyze current evidence regarding bond strength characteristics of 3D-printed resins for permanent/definitive dental restorations, examining various surface treatments, bonding protocols, and aging effects. Understanding these aspects is crucial for establishing evidence-based clinical protocols that ensure the long-term success of 3D-printed permanent restorations.

This meta-analysis seeks to investigate four key research areas: the influence of different surface treatments on bond strength of 3D-printed permanent dental restorations, the effects of various post-processing protocols on bonding performance, the impact of aging conditions on the long-term stability of bonded interfaces, and how bond strength values of 3D-printed restorations compare to conventionally manufactured alternatives. Through systematic evaluation of these factors, this study aims to provide comprehensive guidance for optimizing clinical outcomes with these innovative materials.

## SURVEY METHODOLOGY

### Protocol and registration
This review was conducted according to the Preferred Reporting Items for Systematic Reviews and Meta-Analyses (PRISMA) guideline. The review protocol was not registered.

### Eligibility criteria
Studies were included if they met the following criteria: publication between 2015 to 2024; investigation of 3D printed permanent/definitive dental restorative materials; evaluation of bond strength through standardized testing methods such as shear bond strength; and peer-reviewed articles written in English with clear descriptions of surface treatment protocols and bonding procedures. Studies were excluded if they focused on temporary or provisional 3D-printed materials, did not report quantitative bond strength measurements, or lacked clear description of testing methodology.

### Information sources and search strategy

A comprehensive electronic search was conducted in PubMed, Google Scholar, and Scopus databases. The search terms included combinations of keywords: "3D-printing" or "additive manufacturing" combined with "permanent restoration" or "definitive restoration" and "bond strength" or "shear bond" or "tensile bond" or "push-out bond". Two investigators (M.K. and M.K.) independently performed the search protocol. Any discrepancies were resolved through collective review and discussion of the search terms and context until consensus was achieved.

### Study selection

Three independent investigators screened the titles and abstracts of the identified studies for eligibility. The full texts of potentially relevant studies were retrieved and reviewed independently by the same investigators. Disagreements were resolved through discussion and consensus.

### Data collection process

Data extraction was performed independently using a standardized data extraction form. The extracted information included study identifiers such as author and year of publication, methodological details including the type of 3D-printed material, printer specifications, surface treatment protocols, bonding procedures, and specimen preparation methods. Additional data collected included testing parameters, aging protocols where applicable, and analysis of failure modes.

### Data items

The primary outcome of interest was bond strength, evaluated through various testing methodologies. This included measurements of shear bond strength, microshear bond strength, and push-out bond strength. The review also considered the effects of different surface treatments on bond strength, the impact of aging protocols, and the distribution of failure modes after testing.

### Risk of bias in individual studies

The risk of bias in individual studies was assessed independently by three investigators using the modified CONSORT guidelines for *in vitro* studies. Quality assessment focused on sample size calculation, standardization of specimen preparation, calibration of testing equipment, blinding during measurements, and appropriateness of statistical analysis methods.

### Summary measures and synthesis of results

A qualitative synthesis of the included studies was performed. The extracted data were summarized and organized based on 3D printing materials, bonding materials, testing methodologies, surface treatment protocols, aging conditions, and failure modes. Due to the heterogeneity of testing methodologies and reporting formats across studies, a meta-analysis was not conducted.

### Risk of bias across studies

Publication bias was not assessed due to the limited number of included studies. The quality of evidence was evaluated based on the consistency of findings across studies, precision of bond strength measurements, directness of evidence, and risk of bias in individual studies. This evaluation provided a comprehensive assessment of the reliability and applicability of the findings in the context of clinical relevance.

## RESULTS

### Study selection

The initial electronic database search identified 13 records, with nine studies ultimately meeting all inclusion criteria after applying exclusion parameters (Fig. 1). The included studies, all published between 2023–2024, primarily focused on evaluating bond strength characteristics of 3D-printed permanent dental restorations. Research objectives ranged from comparing different manufacturing methods to examining specific surface treatments and investigating the effects of post-processing protocols. Results were summarized in Tables 1–3.

### Manufacturing methods and study materials

Most studies utilized either digital light processing (DLP) or stereolithography (SLA) technology, with one study examining fused deposition modeling (FDM). Various commercial materials were investigated, including VarseoSmile Crown Plus (Bego), Crowntec (Saremco Dental AG), and Rodin Sculpture series (PacDent). Several studies included control groups using conventional milled materials such as PMMA blocks or machinable resin composites. The selection of printing methods appeared to influence material properties, with DLP and SLA technologies showing comparable results in terms of bond strength potential when proper post-processing protocols were followed. There was one study which did not utilize 3D printing technology, instead fabricating specimens through conventional mold-based techniques with light curing (*Kim et al., 2024*).

### Specimen design and preparation

Specimen geometries varied based on testing methodology, with most studies using standardized shapes including discs (7–10 mm diameter), rectangular bars, and custom forms. For example, *Peskersoy & Oguzhan (2024)* used cylinder shapes (two mm diameter, four mm length), while *Ersöz et al. (2024)* employed rectangular prisms ($12 \times 8 \times 2$ mm) in Table 1. The consistency in specimen dimensions within each study facilitated reliable comparative analyses, though the variation between studies reflected different testing requirements and clinical applications.

### Post-processing protocols

Post-processing emerged as a critical factor in determining final material properties and bond strength outcomes. Washing procedures typically involved isopropyl alcohol or ethanol, with durations ranging from 1 to 10 min in Table 2. *Kagaoan et al. (2024)* specifically demonstrated that extended washing periods could significantly compromise

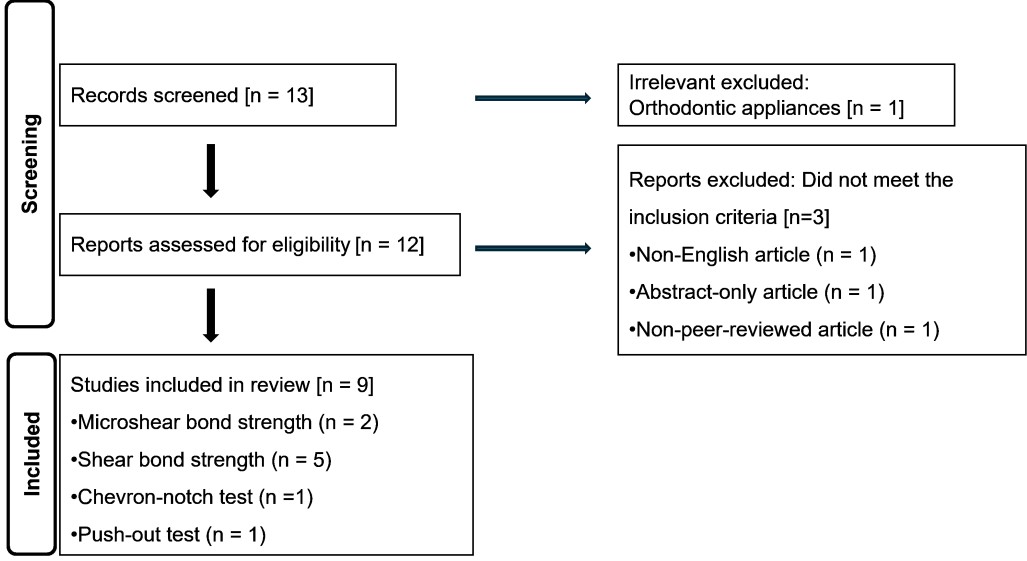

**Figure 1 Flow chart of searching protocol.**

bond strength, emphasizing the importance of adhering to manufacturer-recommended protocols. Post-curing methods showed considerable variation, from 10 min under LED curing to 40 min at 60 °C, reflecting material-specific requirements and manufacturer guidelines.

## Surface treatments and bonding protocols

Surface treatment predominantly involved airborne-particle abrasion, with particle sizes ranging from 25 μm to 110 μm $Al_2O_3$ and pressures from 1.5 to 2.8 bar (0.15 to 0.28 MPa) as seen in Table 2. Studies employed various adhesive systems and cements, with universal adhesives and self-adhesive resin cements being most common. Notable products included Scotchbond Universal (3M), All-Bond Universal (Bisco), and G-Multi Primer (GC). The combination of appropriate surface treatment and bonding protocols proved crucial for achieving optimal bond strength, with several studies demonstrating synergistic effects between sandblasting and specific adhesive systems.

## Thermocycling protocols

Six studies incorporated thermocycling aging protocols, typically implementing 5,000–10,000 cycles between 5 °C and 55 °C. The aging process revealed important differences in material performance, with *Kang et al. (2023)* demonstrating that UDMA-based resins maintained better adhesive stability after thermocycling compared to Bis-EMA-based materials. These findings provided valuable insights into the long-term stability of different material compositions under simulated clinical conditions.

**Table 1  Summary of the bond strengths of 3D-printed permanent/definitive resin; study objective, additive manufacturing method, 3D printing material, 3D printing machine, and design.**

| Author | Objective | Additive manufacturing method | 3D printing material | 3D printing machine | Specimens design |
|---|---|---|---|---|---|
| *Tanaka et al. (2024)* | – Evaluate and compare the mechanical, optical, microstructural, surface, and adhesive behavior of a 3D-printed resin composite and a machinable resin composite | Stereolithography | Vitality (Smart Dent). Control: milling block (Grandio blocks, Voco) | Anycubic Photon Mono (Anycubic) | Disc (10 × 1.5 mm) Bar shape (14.2 × 2.1 × 2.1 mm) |
| *Peskersoy & Oguzhan (2024)* | – Compare the marginal fit of indirect restorations manufactured using 3D printing, CAD/CAM, and traditional indirect composite resin methods. – Compare the bond strength of indirect restorations manufactured using 3D printing, CAD/CAM, and traditional indirect composite resin methods. | Digital light processing | VarseoSmile Crown Plus (Bego). Control: milling blocks (Cerasmart, GC and Signum, Heraeus Kulzer) | Varseo XS (Bego) | Custom onlay (various dimensions). Cylinder shape (2 mm diameter and 4 mm length) |
| *Ersöz et al. (2024)* | – Evaluate the effect of surface treatment methods (sandblasting, hydrofluoric acid, no treatment) on the shear bond strength (SBS) between 3D-printed permanent resins and adhesive cement – Evaluate the effect of 3D printer technology (SLA vs DLP) on the SBS between 3D-printed permanent resins and adhesive cement | Digital light processing Stereolithography | DLP group: Crowntec (Saremco Dental AG). SLA group: Permanent Crown (Formlabs) | DLP group: MAX UV (Asiga). SLA group: Formlabs 3B+ (Formlabs) | Rectangular prism form (12 × 8 × 2 mm) |
| *Kagaoan et al. (2024)* | – Investigate the effect of post-washing duration on the bond strength between additively manufactured crown materials and dental cement – Investigate the effect of crown thickness on the bond strength between additively manufactured crown materials and dental cement | Digital light processing | VarseoSmile Crown Plus (permanent VC; Bego). NextDent C&B MFH (long-term temporary ND; 3D Systems). Control: PMMA block (Telio CAD, Ivovlar Vivadent) | MAX UV (Asiga) | Rectangular bar (15 × 16 × thickness of 1.5 or 2 mm) |
| *Mayinger et al. (2024)* | – Investigate the chemical and mechanical properties of polyphenylene sulfone (PPSU) based on its composition (unfilled or filled with silver-coated zeolites) – Examine how the manufacturing process (granulate, filament, 3D-printed) affects the properties of PPSU | Fused deposition modeling | PPSU1: unfilled PPSUP-PSU2: filled (8% silver coated zeolites) PPSU Specimens were made of granulate (GR), cut from filament (FI), or fabricated by 3D printing (3D) | Apim P155 (Apium Additive Technologies) for PPSU1Apim P220 (Apium Additive Technologies) for PPSU2 | Rectangular shape (2 × 3 × 15 mm) |
| *Kim et al. (2024)* | – Evaluate the shear bond strength of 3D-printed materials for permanent dental restorations – Assess the impact of various surface treatments (e.g., silane, zirconia primer, bonding agent) on the bond strength of these 3D-printed materials | N/S | 3D Printing Resins: Rodin Sculpture 1.0 (PacDent) and Rodin Sculpture 2.0 (PacDent). Control: Aelite All-Purpose Body composite resin (Bisco) | Light-cured by an LED curing unit | Disc (10 mm diameter and 2 mm thickness) |
| *Küden, Batmaz & Karakas (2024)* | – Examine the impact of surface pretreatments commonly employed in conjunction with 3D-printed resin posts on the contact angle (CA), surface free energy (SFE), and push-out bond strength (PBS) | Stereolithography | Permanent Crown Resin (Formlabs) | Form 3 (Formlabs) | Custom post (varied dimensions) |
| *Kang et al. (2023)* | – Identify appropriate adhesion conditions by analyzing shear bond strengths of 3D-printed resins with resin cement based on surface treatment | Group 1: Digital light processing. Group 2: Stereolithography | Group 1: TeraHarz TC-80 (Graphy). Group 2: Permanent Crown Resin (Formlabs) | Group 1: SprintRay Pro 95 (SprintRay). Group 2: Form 3 (Formlabs) | Disc (7 mm in diameter and 3.5 mm in thickness) |
| *Kim et al. (2023)* | – Evaluate the bonding ability of a 3D printing resin and compare it with other indirect resin materials for crown fabrication | Digital light processing | Permanent printing material, Graphy TC-80DP (GP; Graphy). Temporary printing material, Nextdent C&B MFH (NXT; NextDent). Nano-hybrid ceramic, MAZIC Duro (MZ; Vericom). PMMA ceramic, VIPI Block Trilux (VIPI; VIPI) | SprintRay Pro 95 (SprintRay), NextDent 5100 (NextDent) | Disc shape of specimens embedded in the self-polymerizing resin |

## Bond strength testing methods

Bond strength evaluation methods varied across studies, reflecting different aspects of clinical performance requirements shown in Table 3. The most common approach was

**Table 2  Summary of the bond strengths of 3D-printed permanent/definitive resin; washing method, post-curing procedures, airborne-particle abrasives, surface treatment agents and cement, and thermocycling.**

| Author | Washing method | Post-curing procedures | Airborne-particle abrasives | Surface treatment agents and cement | Thermocycling |
|---|---|---|---|---|---|
| *Tanaka et al. (2024)* | Ethanol in an ultrasonic bath for 3 min | EDG Magnabox curing light (EDG Equipamentos) for 10 min (discs) or 20 min (bars) | 50 µm Al2O3 particles at 2 bar for 30 s | Silane: Ceramic Bond. Adhesive: Tetric N-Bond (Ivoclar Vivadent). Cement: Variolink N (Ivoclar Vivadent), Bifix QM (Voco) | 5,000 cycles (5–55 °C) with a dwell time of 60 s |
| *Peskersoy & Oguzhan (2024)* | 96% ethanol for 6 min in a ultrasonic bath | Otoflash unit (Bego), 1,500 flashes at 10 Hz in nitrogen gas for both sides | 30 µm Al2O3 particles at 2.5 bar | Silane: Monobond (Ivoclar Vivadent). Adhesive: Heliobond (Ivoclar Vivadent). Cement: RelyX U200 self-adhesive cement (3M) | 5,000 cycles (5–55 °C), with 20 s dwell time and 10 s transfer time |
| *Ersöz et al. (2024)* | DLP group: 99% isopropyl alcohol for 1 minSLA group: isopropyl alcohol for 3 min | DLP group: Labolight DOU (GC) for 10 minSLA group: FormCure (Formlabs) for 40 min at 60 °C | 50 µm Al2O3 particles at 2.5 bar for 10 s for Group 1 (sandblasting) | Primer: G-Multi Primer (GC). Cement: G-Cem ONE (GC) | N/S |
| *Kagaoan et al. (2024)* | 5 min, 10 min, 1 h, 8 h in 96% ethanol | Permanent VC specimens: Otoflash (Bego) for 2 ×1,500 flashes within protective nitrogen gas atmosphere. Long-term temporary ND specimens: LC 3D Print Box (3D Systems) for 30 min at 60 °C following a preheating period of 15 min | Permanent VC specimens: 50 µm glass beads at 1.5 bar | Adhesive: Monobond Plus (Ivoclar Vivadent) for 3D printing specimens, SR connect (Ivoclar Vivadent) for milled PMMA specimens. Cement: Variolink Esthetic DC cement (Ivoclar Vivadent) | N/S |
| *Mayinger et al. (2024)* | Cleaned in distilled water in an ultrasonic bath | N/S | 110 µm Al2O3 at 0.2 MPa at an angle of 45° from a distance of 10 mm | Adhesive: Adhese Universal (AD; Ivovlar Vivadent), Clearfil Universal Bond Quick (CQ; Kuraray), One Coat 7 Universal (OC; Coltene), Peak Universal Bond (PB; Ultradent), Prime & Bond active (PR; Dentsply Sirona), Scotchbond Universal (SB; 3M), visio.link (VL; bredent). Cement: veneering resin composite (Crea.lign, Bredent), luting resin composite (DuoCem, Coltene) | Thermocycling in 5–55 °C. Martens hardness, elastic indentation modulus and flexural strength specimens for 20 s and bond strength specimens for 30 s in each bath - 5,000 thermal cycles- 10,000 thermocycles - 10,000 thermocycles + 36 days dry storage - 10,000 thermocycles + 36 days dry storage + 10,000 thermocycles |
| *Kim et al. (2024)* | N/S | LC-3DPrint Box (NextDent) for 20 min | 25 µm Al2O3 particles at 2.0 bar for 5 s from 20 cm | Adhesive: All-Bond Universal bonding agent (Bisco). Primer: Porcelain Primer (Bisco). Z-Prime Plus (Bisco). Cement: DuoLink Universal (Bisco) | N/S |
| *Küden, Batmaz & Karakas (2024)* | 99% pure isopropyl alcohol for 5 min | Formcure (Formlabs) at 60 °C for 20 min | 110 µm Al2O3 at 2.8 bar for 30 s from 1 cm | Cement: RelyX U200 (3M) | N/S |
| *Kim et al. (2023)* | 90% isopropyl alcohol for 10 min | Group 1: Cure-M 102H for 20 min. Group 2: Formcure at 60 °C for 30 min | 50 µm Al2O3 at 0.2 MPa for 10 s at 10 mm distance | Adhesive: Single Bond Universal (3M). Cement: Rely X U200 (3M), Rely X Ultimate Cement (3M) | Half of the specimens from each group stored in distilled water for 22–26 h at 37 °C. Other half had 10,000 thermocycles between 5–55 °C, 70 s per cycle) 10,000 cycles. |
| *Kim et al. (2023)* | Washed for 5 min in an ultrasonic washing machine (Twin Tornado, Medifive) with resin cleaner (Twin 3D Cleaner, Medifive) | GP: 30 ×30 min in a post-curing unit (The CureM U102H, Graphy). NXT: 30 min using a post-curing machine (LC-3DPrint box, NextDent) | 50 µm Al2O3 from 10 mm distance, with a pressure of 2 bar, for 10 s | Adhesive: Scotchbond Universal (3M). Cement: Rely X Ultimate (3M) | N/S |

shear bond testing, employed in seven studies using universal testing machines or shear bond tester at a standardized crosshead speed of 0.5 to one mm/min. Among them, two

**Table 3   Summary of the bond strengths of 3D-printed permanent/definitive resin; bond strength test method, debonding classification, and main findings.**

| Author | Bond strength test method | Debonding classification | Main findings |
|---|---|---|---|
| *Tanaka et al. (2024)* | Microshear bond test by a universal testing machine 0.5 mm/min | N/S | - The 3D-printed resin composite had inferior mechanical and optical properties compared to the machinable resin composite.<br>- The 3D-printed resin showed better wear resistance and bond strength, especially after aging.<br>- Improving the distribution of inorganic fillers in the 3D-printed resin is crucial for it to match the performance of the machinable resin.<br>- Bond strength (MPa) of 3D-printed resin<br><br>|  | Bifix QM | Variolink N |<br>|---|---|---|<br>| Initial | 21.76 (6.64) | 26.75 (5.14) |<br>| Aging | 31.90 (12.66) | 24.36 (6.85) |<br><br>- All experimental groups had pretest failures. While printed resin presented more pretest failures in immediately tested groups, machinable resin showed a higher frequency of pretest failures after aging. |
| *Peskersoy & Oguzhan (2024)* | Microshear bond test by a universal testing machine 0.5 mm/min | Adhesive, mixed, cohesive failures | - The marginal fit of indirect restorations produced by 3D printing and subtractive manufacturing (CAD/CAM) were within clinically acceptable levels, with no significant differences between the three fabrication methods.<br>- The CAD/CAM group had the highest bond strength values before and after thermal cycling compared to the 3D printing and indirect composite resin groups.<br>- Mixed failure was the most prevalent fracture pattern, accounting for 60% of cases.<br>- The subtractive manufacturing (CAD/CAM) group had the lowest void volume within the restoration material compared to the 3D printing and indirect composite resin groups.<br>- Bond strength (MPa) of 3D-printed resin: Initial - 12.49 (2.83), After aging - 11.36 (5.41) |
| *Ersöz et al. (2024)* | Shear bond test by a universal testing machine 0.5 mm/min | Adhesive, mixed, or cohesive failures | - Sandblasting created greater surface roughness on 3D-printed samples compared to hydrofluoric acid etching.<br>- Sandblasting groups obtained higher shear bond strength values compared to hydrofluoric acid etching groups.<br>- Cohesive fractures were observed in the sandblasted groups, while mixed fractures were observed in the hydrofluoric acid etching groups.<br>- Bond strength (MPa) of 3D-printed resin<br><br>|  | DLP group | SLA group |<br>|---|---|---|<br>| Sandblasting | 11.6 (1.4) | 9.6 (1.1) |<br>| Hydrofluoric acid | 9.9 (0.8) | 7.4 (1.2) |<br>| No treatment | 8.4 (1.2) | 6.1 (1.3) | |
| *Kagaoan et al. (2024)* | Chevron-notch test by a universal testing machine 0.5 mm/min | Adhesive, mixed, or cohesive failures | - Prolonged post-washing of additively manufactured crown materials in ethanol solution can significantly decrease the bond strength.<br>- Crown thickness does not influence the bond strength of additively manufactured and milled materials.<br>- When post-washed correctly for 5 min, additively manufactured crown materials observed comparable or higher bond strength values compared to PMMA milled crown material.<br>- Bond strength (MPa) of 3D-printed resin<br><br>|  | Permanent VC | | Long-term temporary ND | |<br>|---|---|---|---|---|<br>|  | 5-minute washing | 8-hour washing | 5-minute washing | 8-hour washing |<br>| 1.5 mm thickness | 1.50 (0.94) | 0.81 (0.65) | 1.47 (0.56) | 0.36 (0.24) |<br>| 2.0 mm thickness | 1.32 (0.57) | 0.22 (0.10) | 1.22 (0.87) | 0.43 (0.10) | |

**Table 3** (*continued*)

| Author | Bond strength test method | Debonding classification | Main findings |
|---|---|---|---|
| *Mayinger et al. (2024)* | Shear bond test by a universal testing machine 0.5 mm/min | N/S | - The filled PPSU (PPSU2) specimens showed a homogeneous distribution of the silver-containing zeolite fillers.<br>- The mechanical properties of the filled PPSU (PPSU2) were lower than the unfilled PPSU (PPSU1), with the 3D-printed filled PPSU (PPSU2-3D) exhibiting the lowest flexural strength.<br>- The filled PPSU (PPSU2) specimens exhibited a continuous release of silver ions over a 42-day period, with the 3D-printed filled PPSU (PPSU2-3D) releasing the highest amount of silver.<br>- Shear bond strengths of 3D-printed resin to the luting (7.0–16.2 MPa) and veneering composite (11.8–22.2 MPa), except for adhesive system PR, were shown. |

- Bond strength (MPa) of 3D-printed resin (PPSU2-3D)

| | Veneering resin composite | Luting resin composite |
|---|---|---|
| AD | 21.2 (4.6) | 15.2 (2.1) |
| CQ | 11.8 (3.7) | 15.4 (4.8) |
| OC | 17.8 (3.8) | 11.7 (3.9) |
| PB | 12.8 (3.9) | 7.0 (4.3) |
| PR | 0 | 16.2 (2.0) |
| SB | 22.2 (6.0) | 12.5 (6.0) |
| VL | 19.5 (2.5) | 13.7 (2.0) |

| Author | Bond strength test method | Debonding classification | Main findings |
|---|---|---|---|
| *Kim et al. (2024)* | Shear bond test by a shear bond tester 1 mm/min | Adhesive, mixed, or cohesive failures | - Rodin 1.0 had improved bond strengths with bonding agent application, but increased adhesive failures with just silane or zirconia primer.<br>- Rodin 2.0 had consistent bond strengths regardless of bonding agent, but more cohesive failures with bonding agent and filler coating.<br>- Silane coating increased cohesive failure rates across all groups except Rodin 1.0 without bonding agent. |

- Bond strength (MPa) of 3D-printed resin

| | Without bonding agent | | With bonding agent | |
|---|---|---|---|---|
| | Rodin 1.0 | Rodin 2.0 | Rodin 1.0 | Rodin 2.0 |
| No treatment | 22.93 (6.57) | 39.21 (9.71) | 38.12 (5.89) | 35.36 (6.14) |
| Silane | 23.16 (7.48) | 35.26 (8.40) | 35.08 (5.87) | 35.73 (5.41) |
| ZrO$_2$ primer | 27.48 (8.55) | 36.71 (7.24) | N/A | N/A |

| Author | Bond strength test method | Debonding classification | Main findings |
|---|---|---|---|
| *Küden, Batmaz & Karakas (2024)* | Specimens sectioned into 1 mm thickness at the coronal, middle and apical thirds Push-out test by a universal testing machine 0.5 mm/min | Adhesive, mixed, or cohesive failures | - Silane and sandblasting pretreatments enhanced the surface free energy and push-out bond strength of 3D-printed resin posts.<br>- Hydrogen peroxide and hydrofluoric acid pretreatments did not significantly improve the surface free energy or push-out bond strength of 3D-printed resin posts.<br>- Push-out bond strength values decreased from the cervical to the apical regions of the root canal across all groups. |

- Bond strength (MPa) of 3D-printed resin

| | Glass fiber post (GFC) | Permanent crown resin (PRC) | Sandblasting (SB) | Silane (SL) | Hydrofluoric acid (HF) | Hydrogen peroxide (HP) |
|---|---|---|---|---|---|---|
| Cervical | 3.8 (0.6) | 2.2 (0.7) | 2.6 (0.7) | 3.4 (0.6) | 1.9 (0.6) | 1.8 (0.6) |
| Middle | 2.3 (0.6) | 1.1 (0.4) | 1.4 (0.4) | 2.3 (0.6) | 0.9 (0.4) | 0.8 (0.3) |
| Apical | 1.6 (0.6) | 0.7 (0.3) | 1.3 (0.4) | 1.5 (0.4) | 0.6 (0.4) | 0.3 (0.2) |

- Except in the GFC, SB, and SL groups, mixed failure decreased from the cervical to apical levels, while adhesive failure rates increased.
- GFC cervical showed the highest cohesive failure rate (30%), while HF apical exhibited the highest adhesive failure rate (80%).

| Author | Bond strength test method | Debonding classification | Main findings |
|---|---|---|---|
| *Kang et al. (2023)* | Shear bond test by a universal testing machine 0.5 mm/min | N/S | - The shear bond strength varied depending on the type of 3D printing resin material and adhesion condition.<br>- Combining airborne-particle abrasion and single bond universal treatment resulted in the highest SBS for both resin materials.<br>- The UDMA-based resin (Group 1) showed higher adhesive stability after thermocycling compared to the Bis-EMA-based resin (Group 2). |

- Bond strength (MPa) of 3D-printed resin

| | | CU | AU | CBU | ABU | CBUT | ABUT |
|---|---|---|---|---|---|---|---|
| Group1 | Before thermocycling | 17.0 (3.0) | 20.8 (5.7) | 22.9 (4.2) | 24.7 (4.9) | 21.9 (4.9) | 20.2 (5.2) |
| | After thermocycling | 8.1 (3.1) | 16.9 (4.3) | 11.6 (3.6) | 22.5 (3.3) | 16.8 (3.3) | 17.8 (3.9) |
| Group2 | Before thermocycling | 17.9 (2.8) | 18.2 (2.6) | 15.9 (2.6) | 16.0 (2.8) | 17.3 (2.4) | 19.4 (2.7) |
| | After thermocycling | 12.9 (2.3) | 11.2 (2.6) | 10.8 (1.8) | 11.2 (2.6) | 11.2 (2.3) | 12.9 (3.0) |

![PeerJ]

**Table 3** (*continued*)

| Author | Bond strength test method | Debonding classification | Main findings |
|--------|--------------------------|-------------------------|---------------|
| *Kim et al. (2023)* | Shear bond test by a shear bond tester 1 mm/min | Adhesive, mixed, or cohesive failures | - The 3D-printed resin materials (GP and NXT) and the nano-hybrid ceramic material (MZ) had significantly higher shear bond strengths compared to the organically modified PMMA ceramic material (VIPI).<br>- The 3D-printed resin materials (GP and NXT) and the nano-hybrid ceramic material (MZ) had stronger internal cohesive strength compared to the VIPI material, which exhibited more adhesive and mixed failures. |

- Bond strength (MPa)

| | |
|---|---|
| GP | 23.29 (3.88) |
| NXT | 26.14 (4.67) |
| MZ | 25.41 (4.03) |
| VIPI (PMMA ceramic) | 18.79 (4.26) |

studies utilized microshear bond testing, which allowed for multiple measurements on a single specimen and better stress distribution during testing (*Peskersoy & Oguzhan, 2024*; *Tanaka et al., 2024*). *Kagaoan et al. (2024)* implemented the chevron-notch test, providing insights into crack initiation and propagation characteristics. *Küden, Batmaz & Karakas (2024)* uniquely employed push-out testing by sectioning specimens into one mm thickness at coronal, middle, and apical thirds, enabling evaluation of regional bond strength variations within root canal posts. This diversity in testing methodologies provided complementary perspectives on bonding performance, though it somewhat limited direct numerical comparisons between studies.

## Failure analysis and main findings

The analysis of failure modes provided crucial insights into bonding mechanisms and material behavior. Most studies reported a combination of adhesive, cohesive, and mixed failures, with the distribution pattern often correlating with surface treatment methods and material compositions. Notably, sandblasted specimens generally exhibited more cohesive failures, suggesting enhanced interfacial bonding. Studies comparing 3D-printed materials with conventional alternatives revealed complex relationships between manufacturing methods and bond strength outcomes. While *Tanaka et al. (2024)* reported superior bond strength in 3D-printed resins after aging, particularly in terms of wear resistance, *Peskersoy & Oguzhan (2024)* found higher values in CAD/CAM groups. These apparently contradictory findings highlighted the importance of material-specific optimization and appropriate processing protocols in achieving reliable bonding outcomes. Additionally, the investigation of regional variations in bond strength, particularly in post applications, demonstrated the need for considering application-specific requirements in material selection and processing protocols.

## DISCUSSION

This meta-analysis reveals a complex interplay of factors affecting bond strength in 3D-printed permanent dental restorations, providing insights that extend beyond isolated variables to their integrated effects on clinical performance.

At the core of our findings is the critical relationship between material composition and bonding mechanisms. UDMA-based resins demonstrated not only higher initial bond

strengths (20.8–24.7 MPa) but also maintained superior stability after thermocycling (16.8–22.5 MPa) compared to Bis-EMA alternatives, which declined from 16.0–19.4 MPa to merely 10.8–12.9 MPa after aging. This remarkable stability difference—UDMA retaining up to 91% of its initial strength while Bis-EMA preserved only 62–71%—stems from UDMA's inherent chemical structure providing lower water sorption and higher cross-linking density (*Kerby et al., 2009*). The dramatic contrast between different 3D-printed resin formulations further emphasizes how material composition fundamentally determines bonding potential even when manufacturing processes and surface treatments remain constant. This relationship extends to specialized materials like 3D-printed PPSU resins, where despite decreased mechanical properties due to silver-coated zeolite fillers, clinically acceptable shear bond strengths were maintained. Notably, bond strength values varied considerably depending on bonding agent selection and cementation materials, specifically PR bonding agent did not have any bond with PPSU and veneering resin composite (*Mayinger et al., 2024*). This underscores the importance of not only material composition but also appropriate adhesive system selection for optimizing clinical performance of 3D-printed permanent resins.

Manufacturing technology influences bond strength primarily through its interaction with material composition and post-processing requirements. While both DLP and SLA technologies produced clinically acceptable results, their performance varied significantly based on the specific resin chemistry employed. DLP-manufactured materials showed higher potential with UDMA-based formulations, as evidenced by *Kang et al. (2023)*'s findings with TeraHarz TC-80, while SLA printing demonstrated better aging resistance with certain compositions. This technology-material interrelationship carries important clinical implications for material selection based on specific restoration requirements.

Both mechanical and chemical surface treatments showed variable effectiveness depending on the composition and filler content of 3D-printed resins. Sandblasting with 50 μm aluminum oxide particles consistently enhanced bond strength across materials, with sandblasted specimens achieving significantly higher values than those treated with hydrofluoric acid or left untreated for both DLP (11.6 MPa *vs.* 9.9 MPa *vs.* 8.4 MPa) and SLA materials (9.6 MPa *vs.* 7.4 MPa *vs.* 6.1 MPa). This effectiveness appears to stem from three mechanisms: creation of microretentive features, and exposure of subsurface filler particles that enhance chemical bonding potential (*Li et al., 2018*; *Okada et al., 2019*; *Yoshihara et al., 2017*). Chemical treatments demonstrated material-specific responses, with silane application providing minimal improvement for some materials (22.93 MPa to 23.16 MPa) while zirconia primers yielded better results (27.48 MPa) for the same materials. Notably, higher-filled material maintained consistently high bond strengths regardless of surface treatment (35.26–39.21 MPa with bonding agent and 35.36–35.73 MPa with bonding agent), suggesting material composition sometimes outweighs surface treatment effects in terms of shear bond strengths (*Kim et al., 2024*).

Post-processing protocols significantly influenced bond strength outcomes through their interaction with material composition. Extended washing periods of permanent crown materials drastically reduced bond strength from 1.50 MPa with 5-minute washing to 0.36 MPa after 8 h in ethanol—a 76% reduction that demonstrates the critical importance

of adhering to manufacturer-recommended cleaning procedures (*Kagaoan et al., 2024*). This phenomenon likely results from excessive solvent exposure leaching out unreacted monomers that would otherwise participate in interfacial bonding. Additionally, prolonged washing can decrease the degree of conversion in 3D-printed resins, compromising their mechanical properties and consequently their bonding capability (*Jin et al., 2022*). The interrelationship between washing time, resin chemistry, and mechanical performance underscores the need for standardized, material-specific post-processing protocols to achieve optimal clinical outcomes.

Failure mode analysis across studies revealed significant correlations between bond strength values and failure patterns in 3D-printed permanent resins. Push-out testing demonstrated regional variations in failure modes, with cervical regions showing higher cohesive failure rates while apical areas exhibited predominantly adhesive failures. Except in certain treatment groups, mixed failures generally decreased from cervical to apical regions, while adhesive failure rates increased correspondingly (*Küden, Batmaz & Karakas, 2024*). Higher-strength 3D-printed resins predominantly showed cohesive failures when bonding agents were applied, while lower-strength materials demonstrated more adhesive failures despite various surface treatments, suggesting material composition strongly influences failure patterns (*Kim et al., 2024*). Similarly, higher-strength 3D-printed materials exhibited stronger internal cohesive strength compared to PMMA materials, which showed more adhesive and mixed failures (*Kim et al., 2023*). This pattern suggests that stronger interfacial bonding in 3D-printed resins causes failures to occur within the material itself rather than at the adhesive interface, indicating their potential for stable clinical performance when optimal bond strength is achieved through proper material selection and bonding procedures.

Future research should address several integrated areas: material-specific post-processing optimization that recognizes how washing and curing parameters interact with different resin chemistries; standardized testing methodologies that account for the unique structural characteristics of 3D-printed materials; comprehensive investigations of resin base types, filler compositions, surface treatment protocols, and compatible adhesive systems; and examination of printing parameters like layer thickness and build orientation that create structural variations influencing bonding behavior for permanent/definitive dental 3D-printed restorations. Additionally, testing methodologies should expand beyond conventional bond strength tests to include tensile tests, fatigue loading evaluations, and simulations using natural teeth under oral conditions to better predict clinical performance. Long-term clinical validation studies that extend beyond the current limitations of *in vitro* aging will be essential to confirm the durability of these materials. By addressing these interdependent factors, future studies will provide more comprehensive guidance for optimizing the clinical performance of 3D-printed permanent dental restorations.

## CONCLUSIONS

This meta-analysis demonstrates that 3D-printed permanent dental restorations can achieve clinically acceptable bond strengths when appropriate protocols are followed. Airborne-particle abrasion consistently enhances bond strength, especially when combined with chemical treatments like silane agents and 10-MDP-containing adhesives. Material composition significantly influences bonding performance, with UDMA-based resins showing superior stability after aging compared to Bis-EMA alternatives. Strict adherence to manufacturer-recommended post-processing protocols is essential for optimal bonding outcomes. The predominance of cohesive failures in high-strength materials suggests that bond optimization may ultimately be limited by the inherent strength of the material itself. Long-term clinical studies are necessary to validate the performance of 3D-printed restorations and their bonding to resin cements in the oral environment.

## ACKNOWLEDGEMENTS

The authors acknowledge the use of ChatGPT (version 4o, OpenAI) for assistance in manuscript editing and language refinement. All content was carefully reviewed and verified by the authors.

### Funding
The authors received no funding for this work.

### Competing Interests
The authors declare there are no competing interests.

### Author Contributions
- Minjoo Ki performed the experiments, analyzed the data, performed the computation work, prepared figures and/or tables, authored or reviewed drafts of the article, and approved the final draft.
- Marc Hayashi conceived and designed the experiments, performed the experiments, analyzed the data, prepared figures and/or tables, and approved the final draft.
- Mijoo Kim conceived and designed the experiments, performed the experiments, analyzed the data, performed the computation work, prepared figures and/or tables, authored or reviewed drafts of the article, and approved the final draft.

### Data Availability
   This is a systematic review/meta-analysis.

### Supplemental Information
Supplemental information for this article can be found online at http://dx.doi.org/10.7717/peerj-matsci.35#supplemental-information.

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
