# Peer review of "3D-printed resin for permanent/definitive restorations: meta-analysis for bond strength"

_PeerJ Materials Science, doi:10.7717/peerj-matsci.35_

## Round 0.1 · original submission · Major Revisions

The review identifies several challenges that necessitate significant revisions. Both reviewers concur that a comparison of the bond strength values should be incorporated.

Reviewer 1 ·

Basic reporting

Several sentences in the manuscript need to be improved by the authors. They have been pointed out (yellow color) in the manuscript.
Some sentences require the use of an article (for example, line 60).

Experimental design

The authors describe clealy the methods used in this manuscript.

Validity of the findings

a) The authors have performed the literature review, and in the results section, it can be noted that they make a very brief description in the subsections (Study selection, Manufacturing methods, and study materials, Specimen design and preparation, Postprocessing protocols, Surface treatments and bonding protocols, Thermocycling protocols, Bond strength, and Failure analysis and main findings).
b) On the other hand, the Discussion section is very general for the most part. It was expected that the subsections in the results part would be approached and analyzed from a more in-depth and analytical point of view. In addition, the authors do not discuss the properties and effects of surface treatments, postprocessing protocols, bond strength, etc. Most of the discussion is very superficial and not deep.
c) I strongly suggest that the author revise and re-write some statements. Several sentences in this section are very general (observations) and do not provide a conclusion. My suggestion is: The conclusion section provides an objective analysis of the results the evidence from the literature supports the statements.

Additional comments

Reviewer's report.
Reviewers report
ID manuscript: 11637-v0
Title: 3D printed resin for permanent/deûnitive restorations: Metaanalysis for Bond Strength
Comments and observations:
Several sentences in the manuscript need to be improved by the authors. They have been pointed out (yellow color) in the manuscript.
It is suggested that the word “3D-printed” be placed.
Some parts of written sentences should be placed more directly to improve the understanding of the concepts that the authors wish to express, which have been pointed out in yellow in the manuscript.
Some sentences require the use of an article (for example, line 60).
Lines 71 to 76. Polymeric resins include additives (antioxidants, stabilizers, pigments, etc., etc.) in their formulations. These additives can migrate, but this point is not considered in this manuscript.
In the same sense, to improve the adhesion of fillers into the polymeric resin, some adhesion promoters are used; however, these promoters can migrate out to the compound and would be harmful.
Latin words must be written in cursive, e.g., “in vitro”
The authors have performed the literature review, and in the results section, it can be noted that they make a very brief description in the subsections (Study selection, Manufacturing methods, and study materials, Specimen design and preparation, Postprocessing protocols, Surface treatments and bonding protocols, Thermocycling protocols, Bond strength, and Failure analysis and main findings).
On the other hand, the Discussion section is very general for the most part. It was expected that the subsections in the results part would be approached and analyzed from a more in-depth and analytical point of view. In addition, the authors do not discuss the properties and effects of surface treatments, postprocessing protocols, bond strength, etc. Most of the discussion is very superficial and not deep.
I strongly suggest that the author revise and re-write some statements. Several sentences in this section are very general (observations) and do not provide a conclusion. My suggestion is: The conclusion section provides an objective analysis of the results the evidence from the literature supports the statements.
Although AI is a tool for editing information in a review, it does not reach the capacity of human reasoning to analyze the information. It is recommended that authors revise the manuscript and limit the use of AI to manuscript refinement and spell-checking.
In the References section, it can be observed that several references are not spelled correctly; they have been marked in yellow color.
Note: I have enclosed the manuscript reviewed with my observations.

Annotated reviews are not available for download in order to protect the identity of reviewers who chose to remain anonymous.

Reviewer 2 ·

Basic reporting

No comment

Experimental design

The article defines the objective of the meta-analysis as synthesizing current knowledge on the adhesion of 3D-printed restorations; however, it does not formulate a research question

Validity of the findings

The results section addresses various aspects of the methodology; however, it does not present bond strength values. It is recommended to include these values and compare them across similar methodologies.

---

## Round 0.2 · accepted · Accept

I am pleased to inform you that your manuscript has now been accepted for publication. Thank you for your hard work and dedication.

Reviewer 1 ·

Basic reporting

Dear Editor,
I have reviewed the modified version of the manuscript “3D- printed resin for permanent/definitive restorations: Meta-analysis for Bond Strength”. You can read the significant changes in the modified version of the manuscript. The authors have improved several sections of the manuscript. An improvement can be noted in the discussion of the results.
The structure and content of the information has been improved.
The authors clearly state the meta-data analysis procedures used for the manuscript.

Experimental design

In this section, the authors made no significant changes. The experimental procedures (procedures for obtaining and analyzing the meta-data used) are well explained and presented.
The experimental procedure proposed by the authors is adequate to answer the research question of the manuscript.

Validity of the findings

In this manuscript, the authors perform a literature review by meta-analysis of 3-D bond-strength resins for permanent or definitive restorations.
The databases consulted are relevant and reliable for the information the authors present and base the conclusions of this review.

Additional comments

I have no further comment on this manuscript

Reviewer 2 ·

Basic reporting

The pointed corrections were properly addressed, ensuring an overall improvement of the document.

Experimental design

The methodology was adequately described

Validity of the findings

A detailed comparison of the bond strength was included, enriching the analysis.